# Discrimination of *Bacillus cereus* Group Members by MALDI-TOF Mass Spectrometry

**DOI:** 10.3390/microorganisms9061202

**Published:** 2021-06-02

**Authors:** Viviana Manzulli, Valeria Rondinone, Alessandro Buchicchio, Luigina Serrecchia, Dora Cipolletta, Antonio Fasanella, Antonio Parisi, Laura Difato, Michela Iatarola, Angela Aceti, Elena Poppa, Francesco Tolve, Lorenzo Pace, Fiorenza Petruzzi, Ines Della Rovere, Donato Antonio Raele, Laura Del Sambro, Luigi Giangrossi, Domenico Galante

**Affiliations:** 1Istituto Zooprofilattico Sperimentale della Puglia e della Basilicata, Via Manfredonia 20, 71121 Foggia, Italy; viviana.manzulli@izspb.it (V.M.); luigina.serrecchia@izspb.it (L.S.); dora.cipolletta@izspb.it (D.C.); antonio.fasanella@izspb.it (A.F.); antonio.parisi@izspb.it (A.P.); laura.difato@izspb.it (L.D.); michela.iatarola@izspb.it (M.I.); angela.aceti@izspb.it (A.A.); elena.poppa@izspb.it (E.P.); francesco.tolve@izspb.it (F.T.); lorenzo.pace@izspb.it (L.P.); fiorenza.petruzzi@izspb.it (F.P.); ines.dellarovere@izspb.it (I.D.R.); donatoantonio.raele@izspb.it (D.A.R.); laura.delsambro@izspb.it (L.D.S.); luigi.giangrossi@izspb.it (L.G.); domenico.galante@izspb.it (D.G.); 2Bruker Italia s.r.l., Daltonics Division, Strada Cluentina, 26/R, 62100 Macerata, Italy; a.buchicchio@bruker.com

**Keywords:** *Bacillus anthracis*, *Bacillus cereus* group, MALDI-TOF, mass spectrometry

## Abstract

Matrix-Assisted Laser Desorption/Ionization Time Of Flight Mass Spectrometry (MALDI-TOF MS) technology is currently increasingly used in diagnostic laboratories as a cost effective, rapid and reliable routine technique for the identification and typing of microorganisms. In this study, we used MALDI-TOF MS to analyze a collection of 160 strains belonging to the *Bacillus cereus* group (57 *B. anthracis*, 49 *B. cereus*, 1 *B. mycoides*, 18 *B. wiedmannii*, 27 *B. thuringiensis*, 7 *B. toyonensis* and 1 *B. weihenstephanensis*) and to detect specific biomarkers which would allow an unequivocal identification. The Main Spectra Profiles (MSPs) were added to an in-house reference library, expanding the current commercial library which does not include *B. toyonensis* and *B. wiedmannii* mass spectra. The obtained mass spectra were statistically compared by Principal Component Analysis (PCA) that revealed seven different clusters. Moreover, for the identification purpose, were generated dedicate algorithms for a rapid and automatic detection of characteristic ion peaks after the mass spectra acquisition. The presence of specific biomarkers can be used to differentiate strains within the *B. cereus* group and to make a reliable identification of *Bacillus anthracis*, etiologic agent of anthrax, which is the most pathogenic and feared bacterium of the group. This could offer a critical time advantage for the diagnosis and for the clinical management of human anthrax even in case of bioterror attacks.

## 1. Introduction

The *Bacillus cereus* group is comprised of Gram-positive, rod-shaped and spore-forming aerobic bacteria, genetically closely related. Currently this group includes several species, phylogenetically organized in three broad clades: *Bacillus cereus* sensu stricto and *Bacillus thuringiensis*, both found in all clades; *Bacillus anthracis* and *Bacillus wiedmannii*, in Clade 1; *Bacillus mycoides*, *Bacillus pseudomycoides*, *Bacillus weihenstephanensis*, *Bacillus toyonensis*, *Bacillus cytotoxicus*, *Bacillus bingmayongensis*, *Bacillus gaemokensis* and *Bacillus manliponensis*, in Clade 3 [1,2]. Clades are divided into seven different subgroups, defined according to multiple and combined genetic analyses [2]. 

Despite being extremely similar and related, only some of these microorganisms have an important impact on human and animal health, agriculture and the food industry [3]. 

*B. cereus* sensu stricto is a ubiquitous bacterium, widespread in the terrestrial and marine environment. Thanks to ability to produce spores, it survives pasteurization [4] and the various thermal processes used in food industries [5]; therefore, it is not uncommon to isolate it from both raw and cooked foods. It can produce toxins, causing two types of gastrointestinal illness: the emetic syndrome (associated with nausea and vomiting) and the diarrhoeal syndrome [6,7]. 

*B. anthracis* is the etiological agent of anthrax, an acute fatal zoonotic disease that affects primarily herbivores, but that may be transmitted also to humans. It is also known for its potential use as biological weapon. Compared to the other members of the *B. cereus* group, the genome consists of the chromosome and of two plasmids, pXO1 and pXO2, which carry genes coding for the toxins and the capsule, respectively, and on which its high virulence depends [8]. 

In the past few decades, two atypical *B. cereus* strains (called *Bacillus anthracis*-like bacteria) have been described; they possess *B. cereus* s.s. chromosomal DNA and virulence plasmids that are highly comparable to the anthrax virulence plasmids (pXO1 and pXO2). As they have the same pathogenicity as *B. anthracis* with the production of the tripartite toxin, these strains cause fatal anthrax-like disease in humans and animals [2]. 

Being genetically and phenotypically very similar to each other since they probably have common evolutionary relationships [1], the discrimination of members belonging to the *B. cereus* group is complicated.

The classic methods for microorganism identification are primarily based on biochemical tests, which are very time-consuming with results that are often ambiguous and not exhaustive, and biomolecular tests that are faster but requiring the use of specific reagents (as a set of suitable primers), not always available for routine laboratories diagnostics.

A powerful, sensitive and reliable technique now increasingly used for microbial identification and clinical diagnosis is MALDI-TOF mass spectrometry. Through this valuable diagnostic tool, the discrimination of microbial species different from the genetic and protein point of view is relatively simple but the distinction of closely related species is the real challenge.

Several studies have been conducted already on the applicability of MALDI-TOF MS in the identification/discrimination of related species, as for members of the *B. cereus* group [9,10,11,12]. 

In this study, a collection of 160 strains belonging to the *Bacillus cereus* group was analyzed with the aim to define the characteristic ion peaks for the species *B. cereus*, *B. anthracis*, *B. mycoides*, *B. thuringiensis* and *B. weihenstephanensis*, and create an algorithm to identify the species after the classical library matching approach. This method is already used by the additional software module MALDI Biotyper (MBT) Compass Subtyping for the correct species identification for *Listeria* [13]. The *Listeria* species have very similar spectra profile, and the classical identification approach allows the genus identification, but a new algorithm developed by Bruker Daltonik is able to identify correctly the species using a few characteristic peaks [14].

In addition, the study has been extended also to other species of the *Bacillus cereus* group such as *B. toyonensis* and *B. wiedmannii*, since, presently, they are not included in the Bruker Daltonik (BDAL) library (Bruker Daltonik GmbH, Bremen, Germany).

## 2. Materials and Methods

### 2.1. Bacterial Strains and Molecular Identification 

Bacterial strains used in this study included *n* = 103 strains isolated from food samples, of which *n* = 49 *B. cereus*, *n* = 1 *B. mycoides*, *n* = 18 *B. wiedmannii*, *n* = 27 *B. thuringiensis*, *n* = 7 *B. toyonensis*, *n* = 1 *B. weihenstephanensis* and *n* = 57 *B. anthracis* strains, different for MLVA 31-loci profile, isolated from 1984 to 2020 from animal anthrax outbreaks in Italy and belonging to the collection of the Anthrax Reference Institute of Italy (Ce.R.N.A.). The complete list of isolates used in this study and their origin are listed in Appendix A.

Bacterial isolation from food was performed according to ISO 21871:2006. Briefly, 5 g or mL of sample were added to 45 mL of Buffered Peptone Water (BPW) (Biolife Italiana, Milan, Italy) in sterile bags and homogenized using a stomacher (PBI International, Milan, Italy) at 230 rpm for 30 s. Then, 45 mL of double-strength Tryptone Soy Polymyxin Broth (TSPB) (Biolife Italiana, Milan, Italy) were added to initial suspension and the bags were incubated at 30 °C for 48 h under aerobic condition.

After incubation, 10 mL of enrichment broth were streaked onto the surface of solid selective medium Mannitol Egg Yolk Polymyxin Agar (MYP) (Biolife Italiana, Milan, Italy) and the plates were incubated at 30 °C for 24–48 h under aerobic condition. 

After this step, typical presumptive *Bacillus cereus* group colonies for each sample were picked, subcultured on Columbia Agar with 5% sheep blood and then incubated at 37 °C for 18–24 h. Bacterial DNA was extracted using the DNAeasy Blood and Tissue kit (Qiagen, Hilden, Germany) following the manufacturer’s protocol for Gram-positive bacteria.

Biomolecular identification of *B*. *anthracis* was performed using Real-time PCR assay previously described by Wielinga et al. (2011) [15], based on the amplification of three different specific sequences: *pl3* gene located on chromosome; *pagA* gene located on the virulence plasmid pXO1 and *capB* gene located on the virulence plasmid pXO2. 

Two combined assays were used for the identification of *B. cereus s.s.*, *B. mycoides*, *B. thuringiensis* and *B. weihenstephanensis*: a multiplex PCR based on the *gyrB* sequence [16] and a TaqMan assay based on the amplification of the *motB* gene [17]. 

A TaqMan assay, designed on a specific region of the *ccpA* gene [18], was used to identify *B. toyonensis.*

Lastly, biomolecular identification of *B. wiedmannii* was carried out by 16S rRNA gene sequencing [19].

The manipulation of *B*. *anthracis* strains was performed in a biosafety level 3 (BSL-3) laboratory within a class II safety cabinet.

### 2.2. Sample Preparation for MALDI-TOF Mass Spectrometry Analysis 

Each *Bacillus* strain was grown on Columbia blood agar for 18–24 h at 37 °C before MALDI-TOF MS analysis. For the inactivation of vegetative cells and spores of *Bacillus* species, trifluoroacetic acid (TFA) (Sigma-Aldrich, St Louis, MO, USA) was used, an organic solvent that effectively inactivates the vegetative and spore forms and solubilizes bacterial proteins, preserving their structural integrity [20].

Using a 10 μL loop, fresh bacterial colonies were transferred into a tube containing 1 mL of 80% TFA, resuspended and incubated for 30 min at room temperature. Subsequently, a 1:2 dilution in sterile deionized water (Carlo Erba Reagents, Cornaredo, Italy) was prepared [11].

Thus, 1 μL of this mixture was applied onto a 96-well steel target plate (Bruker Daltonics, Germany). After drying, the sample spots were overlaid with 1 μL of matrix solution, α-cyano-4-hydroxycynnamic acid (HCCA, Bruker Daltonik GmbH, Bremen, Germany) 10 mg/mL. The mass spectra were acquired using Microflex LT/SH™ mass spectrometer (Bruker Daltonik GmbH, Bremen, Germany), which was operated in linear positive mode covering a mass to charge ratio (*m*/*z*) between 2000 and 20,000. Each spot of the target plate was hit with a pulsed nitrogen laser beam operating at 337 nm, with a frequency equal to 60 Hz. After laser shot, the gas phase ions obtained were accelerated in the flight tube by an acceleration voltage with optimized values for the mass range understudy. Two hundred and forty individual laser shots were added for each spectrum. For each strain, a total of 18 mass spectra were obtained. The instrument was calibrated in the broad molecular weight range between 2 and 20 kDa using Bruker Bacterial Test Standard (BTS, Bruker Daltonik GmbH, Bremen, Germany), an extract of the Escherichia coli DH5α strain, with the addition of two proteins (RNase A of 13,683.2 Da and myoglobin of 16,952.3 Da).

### 2.3. Data Processing

The data were processed automatically by MBT Compass 4.1.70 software (Bruker Daltonik GmbH, Bremen, Germany) and the mass spectra were compared with those of known microbial isolates of the commercial libraries provided by Bruker Daltonik, which currently contains about 9000 reference bacterial protein mass spectra (MBT Compass library v 7.0.0.0), including the Security Library (SR). Regarding the *B. cereus* group, the BDAL library includes the following species: *B. cereus* sensu stricto (*n* = 4), *B. thuringiensis* (*n* = 1), *B. mycoides* (*n* = 1), *B. pseudomycoides* (*n* = 1), *B. weihenstephanensis* (*n* = 1), *B. cytotoxicus* (*n* = 4). The SR library clearly includes only *B. anthracis* (*n* = 23). 

At the moment, the BDAL library does not include *B. toyonensis* and *B. wiedmannii* but the MALDI Biotyper system offers the opportunity to expand the library.

About this, two strains of *B. toyonensis* and *B. weidmanni* were cultured and prepared by the full extraction protocol procedures according to the manufacturer’s protocol. Four colonies were transferred in 300 µL of sterile deionized water (Carlo Erba Reagents, Cornaredo, Italy) in a tube, and 900 µL of ethanol absolute (Sigma-Aldrich, St Louis, MO, USA) were added. After vortex mixing and centrifuge, the supernatant was discarded; the dry pellet was resuspended in 25 µL of 70% formic acid (Carlo Erba Reagents, Cornaredo, Italy) and 25 µL of acetonitrile, grade HPLC (Honeywell, Charlotte, North Carolina, United States). After centrifugation, 1 µL of solution was spotted on each spot of the target plate and covered by HCCA matrix after solvent evaporation. For each extract, eight different spots were prepared on the 96-well steel target plate and three different mass spectra were acquired for each of them automatically, thanks to Automatic execute run by Flex Control software (Bruker Daltonik GmbH, Bremen, Germany). 

The mass spectra obtained were manually analyzed by FlexAnalysis software (v 3.4; Bruker Daltonik GmbH, Bremen, Germany) and each spectrum was subjected to spectral preprocessing procedures, such as smoothing, baseline subtraction and intensity normalization. The used parameters were the same as the default MBT processing method, applied for the automatic identification: smoothing alghoritm SavitzkyGolay, Baseline substraction alghoritm TopHat.

Twenty-four different mass spectra for species were analyzed by Flex Analysis software to verify the accuracy in terms of mass to charge ratio and the reproducibility. All the 24 mass spectra acquired for each strain were compared to verify the presence of flatline spectra or spectra with outlier peaks. Furthermore, the mass accuracy inter-spectra were evaluated by checking the mass shift of the base peak in all the acquired mass spectra, considering as maximal tolerance 500 ppm. After the spectra quality check, in terms of reproducibility and accuracy, the new MSPs for *B. toyonensis* and *B. weidmannii* were created by MBT Compass Explorer module (Bruker Daltonik GmbH, Bremen, Germany). The MSPs were created using the Biotyper Standard Method for the spectra processing and the reference peak list creation, to obtain comparable data also with reference mass spectra of BDAL library. In particular, 3000 *m*/*z* as lower bound, 15,000 as upper bound, 25% as minimum desired peak frequency, 70% as maximum desired peak number for MSP, were set as parameters for MSP creation.

The new MSPs were tested preparing fresh subculture of the same strains, for an additional quality control step, and later measured against the newly created MSP. The strains were identified with the new MSP appearing within the 10 best matches of analysis results.

All the experimental mass spectra for all the species were processed together in ClinProTools software (v. 3.0, Bruker Daltonik GmbH, Bremen, Germany) in all mass range acquired (2–20 kDa) with topHat baseline substraction (10% minimal baseline width), SavitzkyGolay smoothing (2.0 width *m*/*z*) and all peaks with signal-to-noise ratio higher than 3 were evaluated. All these parameters were considered for all following statistical methods (Gel view, PCA, average spectra calculation).

The Gel View displays all mass spectra of the loaded classes arranged in a pseudo-gel like look. The *x*-axis records the *m*/*z* value. The left *y*-axis displays the running spectrum number originating from subsequent spectra loading. The peak intensity is expressed by a color code. The color bar and the right *y*-axis indicate the relation between the color a peak is displayed with and the peak intensity in arbitrary units. In the next figure of gel view, the peak intensities are shown as gray scaled.

ClinProTools offers a statistical data analysis in terms of principal component analysis (PCA). The scaling method used was the Level Scaling, where the intensity of each peak is taken in count as well the mean intensity of the peak in the data set.

From the preprocessed individual mass spectra for each species, an average spectrum is calculated. Mass spectra are weighted with the reciprocal size of the classes to get an equal representation of classes with a quite different number of mass spectra.

The peak picking to create the average spectrum for each species were performed using the total average spectrum peak picking approach. Thereby, the peak picking is applied on the calculated total average spectrum. 

The automatic detection of ion peaks is based on the analysis of a smoothed first derivative. The smoothing is determined by the Resolution parameter fixed as appropriate value. To reduce the number of ion peaks picked on the total average spectrum, and thus the average ion peak list, a Signal to Noise Threshold 3.0 was applied and a Maximal Peak Number found value was fixed to 100.

ClinProTools supports also different algorithms for generating classification models. In this study, the classification models were generated using the algorithms QC (Quick Classifiers) and SVM (Support Vector Machine). Usually, to evaluate the performance of the classification models the Recognition Capability and the Cross-Validation parameters are considered. The first is calculated as the relative number of correct classified data points by the classifier for the given model. The second is a measure for the reliability of a calculated model and can be used to predict how a model will behave in the future. Both parameters were calculated for overall data and for each class considered, in our case the different species of *Bacillus*.

## 3. Results and Discussion

For the MALDI-TOF MS analysis, the results are generally expressed with log(score) values between 0 and 3.0, indicative of the matching between the sample spectrum and the MSPs in the reference database. A log(score) <1.7 indicates that it could not identify the genus or species of the strain; a log(score) between 1.7 and 2.0 indicates that identification could be reliable only at the genus level, while a log(score) ≥2.0 indicates that identification could be reliable at the species level of the organism. 

In this study, as expected, the identification with commercial databases proved to be inconclusive for the *B. cereus* group species because almost all samples mass spectra matched with *B. anthracis* (in the SR), in most cases, with a log(score) >2.0 (data not shown). Excluding the SR library, the mass spectra of the samples corresponded to those of the *B. cereus* of the library BDAL with a log(score) generally between 1.7 and 2.0 or the identification was not possible (Data not shown). Furthermore, when one of the *Bacillus cereus* group members is identified by the MBT Compass (Bruker Daltonik GmbH, Bremen, Germany) software with BDAL library, the comment appears: “*Bacillus anthracis*, *cereus*, *mycoides*, *pseudomycoides*, *thuringiensis* and *weihenstephanensis* are closely related and members of the *Bacillus cereus* group. In particular, *Bacillus cereus* spectra are very similar to spectra from *Bacillus anthracis*. *Bacillus anthracis* is not included in the MALDI Biotyper database. For differentiation, an adequate identification method must be selected by an experienced professional. The quality of spectra (score) depends on the degree of sporulation: Use fresh material”. It means that after the identification, additional tests are mandatory to correctly discriminate and identify these species due to the high degree of similarity of the mass spectra. In addition, not all the species belonging to the *Bacillus cereus* group are included in the library, such as *B. toyonensis* and *B. weidmannii.* At this scope, well-characterized strains of these bacteria were used to create new MSPs to add to the BDAL library. Differently from similar previous studies, we also tested the two above mentioned bacteria, in order to implement a new library and to allow the identification of these species, not possible at the moment, even if with some limitations.

Unfortunately, the main spectra profiles for *B. toyonensis* and *B. weidmannii* are enough different from the MSPs of other species contained in the library but too much similar between them, consequently a further algorithm is needed.

The same approach used to discriminate the species *B. cereus*, *B. anthracis*, *B. thuringensis* and *B.*
*weihenstephanensis*, was used also to discriminate *B. toyonensis* from *B. weidmannii*.

The comparison of unknown and reference mass spectra involves the analysis of the total spectrum and comparison of only distinctive peaks. The analysis of specific peaks is more selective, because it “weighs” peaks specific for a given microorganism, excluding those originating from background noise or exogenous factors. Therefore, to investigate the differences at the single peak level, it is mandatory for a detailed analysis for all peaks to be revealed in the mass spectra.

The mass spectra of these species resulted to be differentiable based on their overall peak number, consistent for each *Bacillus cereus* group species (Figure 1).

Mass spectra obtained from each single tested species were compared to each other and for each species a main spectrum (MSP) was created and included in an in-house reference library.

In the dendrogram generated by MSPs (Appendix A) by MBT Compass Explorer software (v. 4.1.70, Bruker Daltonik GmbH, Bremen, Germany), *B. anthracis* protein mass spectra were placed on a separate branch compared to the branch containing most of the protein mass spectra of the remaining *Bacillus* species, except for some *Bacillus cereus* strains that shared the same branch. 

All mass spectra were processed also with ClinProTools software (v. 3.0, Bruker Daltonik GmbH, Bremen, Germany) in all mass range acquired (2–20 kDa) with topHat baseline substraction (10% minimal baseline width), SavitzkyGolay smoothing (2.0 width *m*/*z*) and all peaks with signal-to-noise ratio higher than 3 were evaluated.

ClinProTools software (v. 3.0, Bruker Daltonik GmbH, Bremen, Germany) allowed the generation of simulated gel views (Figure 2) from preprocessed mass spectra, in order to give an overview of the analyzed microbial MALDI mass spectra. These gel views show spectral peak intensities gray scaled as function the *m*/*z* values. Analysis of the gel view demonstrated a high degree of spectral reproducibility between different mass spectra of the same species.

Mass spectra, grouped for each *Bacillus cereus* group species, were statistically compared using Principal Component Analysis (PCA) to determine general differences in the protein mass spectra, by ClinProTools software (v. 3.0, Bruker Daltonik GmbH, Bremen, Germany). Analysis by PCA revealed seven different clusters (Figure 3).

The clusterization obtained by the PCA means that mass spectra for different species can be differentiated by statistical analysis, and therefore mass spectra will necessarily show spectral differences and probably characteristic peaks.

The software generated a list of the 100 ion peaks considered with their mass to charge ratio, the difference between the maximal and the minimal average peak intensity of all classes (Dave), *p*-value of *t*-test, *p*-value of Wilcoxon test, *p*-value of Anderson-Darling test, the Peak intensity average for each class and relative standard deviation and percentage coefficient of variation. To reduce the data to a shorter list, only the mass peaks with higher Dave and the lower standard deviation for the peak intensity average of the main class were considered. A shorter list of twelve ion peaks was obtained (Table 1). 

These twelve ion peaks seem to be characteristic for five out of seven considered species. In particular, from this peak selection, no characteristic ion peaks appear for *B. cereus* and *B. toyonensis*.

For *B. anthracis, B. thuringensis* and *B. weinsthephanensis* species, more than one characteristic ion peak was defined. In particular, for *B. anthracis* the ion peaks 3339 *m*/*z*, 3592 *m*/*z*, 4871 **m*/*z*,* 9740 *m*/*z*; for *B. thuringensis* the ion peaks 2956 *m*/*z*, 2968 *m*/*z*, 3411 *m*/*z*; for *B. weinsthephanensis* the ion peaks 4637 *m*/*z*, 7324 *m*/*z* 9272, can be considered as characteristic ion peaks. This allowed to create 2D Peak Distribution View (Figure 4) showing a perfect separation of the considered species; each colorful mark represents one mass spectra and, the value for x and y, the intensities for the peak reported on the axis.

The four characteristic ion peaks for *B. anthracis* are reported in two different plots: in the first is shown the intensity of the 3339 *m*/*z* against the intensity of the 4871 *m*/*z*; in the second the 3592 *m*/*z* vs. 9740 *m*/*z*. For the other species, the two most intense peaks of the three were considered (Figure 4). These 2D distribution are the first confirmation of the presence of characteristic ion peaks in the mass spectra analyzed for the species considered, useful for species differentiation in the *B. cereus* group.

The same statistical approach was used for *B. mycoides* and *B. wiedmannii* and only one characteristic ion peak was found, in the mass range between 5410 and 5450 as mass-to-charge ratio; the ion peaks selected well defined the two species (Figure 5).

The twelve characteristic ion peaks considered in this study were forced in statistical model to “classify” our seven different species of *B. cereus* sensu lato to describe the mass spectra of the model generation classes in such a way that new mass spectra can be classified afterwards.

The results for the two classifying model are summarized in Table 2.

The recognition capability and the cross-validation values were considered an index of algorithm functionality. The recognition capability of the QC was 68.3%, while cross validation was 66.59%. The recognition capability of the SVM was 97.14%, while cross validation was 96.02%. These values take into account two classes with no characteristic ion peaks as *B. cereus* and *B. toyonensis,* which reduces the percentage values. Nevertheless, the results of the classification model showed an extremely high reliability, especially for the species in which one or more characteristic ion peaks are considered. 

In conclusion, MALDI-TOF mass spectra of *B. thuringiensis* exhibited specific signals at 2956, 2968, 3411 Da; *B. mycoides* specific signals at 5422 Da; *B. anthracis* at 3339, 3592, 4871, 9740 Da; *B. weihenstephanensis* at 4637, 7324, 9272 Da; and *B wiedmannii* exhibited specific signals at 5443 Da.

All this information was used to create a dedicated algorithm in FlexAnalysis software (v. 3.4, Bruker Daltonik GmbH, Bremen, Germany) for a rapid and automatic detection of characteristic ion peaks after the acquisition for the identification purpose. This software has an interesting tool named Specification check. Two different sets of algorithms were created: one useful after the identification results as *B. cereus* group to discriminate *B. cereus*, *B. anthracis*, *B. mycoides*, *B. thuringensis* and *B. weisthephanensis*, and a second algorithm to distinguish *B. toyonensis* and *B. weidmannii*. The algorithm creates the ion peak list of the experimental spectrum acquired after the automatic identification. For each species under study, a list of characteristics ion peaks was created. The algorithms search the characteristic ion peaks with a defined error in terms of mass accuracy and report the ion peaks found in the experimental spectrum. If we find ion peaks considered characteristic for a certain species, the differentiation at species level for *B. cereus* group is considered satisfactory.

In Figure 6, the result obtained after the automatic algorithm run in FlexAnalysis software (v. 3.4, Bruker Daltonik GmbH, Bremen, Germany) is shown. It refers to the algorithm used for *B. anthracis* species, the more complex one, due to the number of the ion peaks. In the example, all the four characteristic ion peaks were found with a low deviation error in term of ppm; in fact, in the result, “Specification passed” is shown. It means that the experimental spectrum can be identified now as *B. anthracis* with no limitations as before.

If all the characteristic ion peaks are found in the experimental spectrum considered, the specification is passed and the correct identification at species level is possible.

When all the other algorithms were applied to the same experimental spectrum, no characteristic ion peaks for other species were found. In fact, the result appears as “NOT PASSED” in Figure 7.

It means that the experimental spectrum can be identified now as *B. anthracis* with no limitations as before.

## 4. Conclusions

Members of the *Bacillus cereus* group are strongly genetically related [21]; therefore, their discrimination is difficult. Developing fast, extremely sensitive and reliable protocols for their detection is extremely important, since some of the species belonging to this group have an important impact on human and animal health [3]. Early identification of *B. anthracis* is crucial, as it could offer a critical time advantage in the management of animal anthrax outbreaks and of clinical treatment of human anthrax, including bioterrorist attacks.

MALDI-TOF mass spectrometry is a valuable tool to rapidly identify and characterize microorganisms isolated from clinical and environmental samples, with results often more reliable than classic microbiological diagnostic methods [22]. 

This analytical technique is based on the ionization of high molecular weight biological macromolecules and separation of the ions, based on their mass to charge ratio (*m*/*z*) [23]. The obtained mass spectra appear as a set of ion peaks of different intensity, each corresponding to the mass/charge value of a molecular ion [23]. 

Microbial identification is possible because each bacterial species is characterized by a specific protein pattern and therefore by a characteristic spectral profile which constitutes its “fingerprint” [24]. The proteins play an important role in laboratory diagnostics as they constitute about 50% of the dry weight of vegetative bacterial cells and are present in multiple copies on the contrary DNA [25]; moreover, they provide good signals bypassing extraction or amplification steps.

Despite the high diagnostic potential, its limit is the fact that microorganisms belonging to different genera have quite different mass spectra with few or no common ion peaks and therefore are easily identifiable [26]. At the species level, mass spectra are increasingly similar [26] and distinguishing them is more complicated, especially with closely related species, as members of the *B. cereus* group. To overcome this issue, in this study, we examined a collection of 160 strains belonging to the *Bacillus cereus* group, isolated from various clinical, food and environmental matrices. The experimental mass spectra obtained with the Flex Control (v. 3.4) and MBT Compass (v. 4.1.70), were then analyzed by the MBT Compass Explorer (v. 4.1.70), ClinProTools (v. 3.0) and FlexAnalysis (v. 3.4) softwares by Bruker Daltonik GmbH, Bremen, Germany. 

The MSPs obtained were added to an in-house reference library, expanding the commercial BDAL library (Bruker Daltonik GmbH, Bremen, Germany), which currently does not include *B. toyonensis* and *B. wiedmannii* mass spectra. As described by previous studies [11,27], a richer database can remarkably increase the probability of a correct identification. In case of unambiguous or doubt identification, however, we recommend to combine this diagnostic method with others, such as biomolecular techniques.

The mass spectra grouped for each *Bacillus cereus* group species were statistically compared using Principal Component Analysis (PCA) to determine general differences in the protein spectra, through which seven different clusters were revealed. 

From the preprocessed individual mass spectra for each species, a total average spectrum was calculated, and, for all classes/species, characteristic ion peaks have been identified. This information has been used to generate two different sets of algorithms for a rapid and automatic detection after the mass spectra acquisition. In particular, the first algorithm was useful to discriminate *B. cereus*, *B. anthracis*, *B. mycoides*, *B. thuringensis* and *B. weisthephanensis*, while the second one was used to distinguish *B. toyonensis* and *B. weidmannii*. 

This study aims to provide an important diagnostic support in the rapid identification of species belonging to a group of highly correlated bacteria, such as the *Bacillus cereus* group, with particular attention to the most pathogenic and feared bacterium of the group, *B. anthracis*, for which the classical diagnostic methods can take a long time, often with ambiguous results. 

## Figures and Tables

**Figure 1 microorganisms-09-01202-f001:**
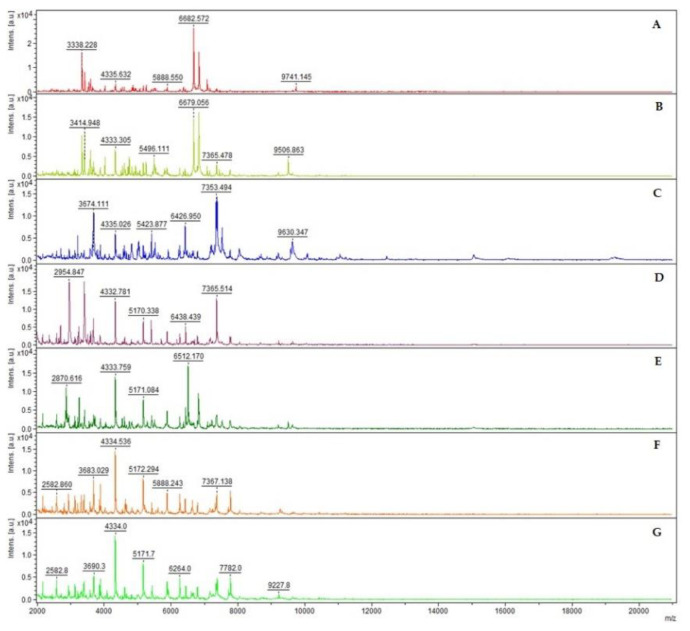
MALDI-TOF mass spectra of seven different *Bacillus* species in the mass range of 2 to 20 kDa: *B. anthracis* (**A**), *B. cereus* (**B**), *B. mycoides* (**C**), *B. thuringiensis* (**D**), *B. toyonensis* (**E**), *B. weihenstephanensis* (**F**) and *B. wiedmannii* (**G**). The mass spectra were obtained using the FlexAnalysis software (v. 3.4, Bruker Daltonik GmbH, Bremen, Germany), performing baseline corrected, smoothed and normalized analyses. The mass spectra demonstrate a relatively high signal-to-noise ratio, which typically permits the detection of 50 to 100 mass peaks per spectrum with the signal-to-noise ratio higher than 3.

**Figure 2 microorganisms-09-01202-f002:**
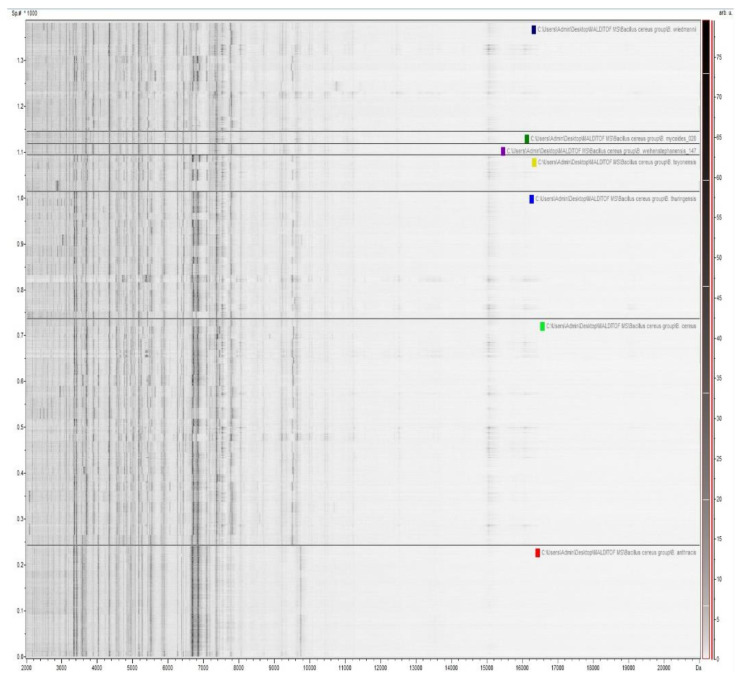
Gel view representation of mass spectra from the *B. cereus* group strains in the mass range of 2 to 20 kDa, tested in this study. The intensities of MALDI-TOF mass spectra are grey scaled and plotted as functions of the *m*/*z* values.

**Figure 3 microorganisms-09-01202-f003:**
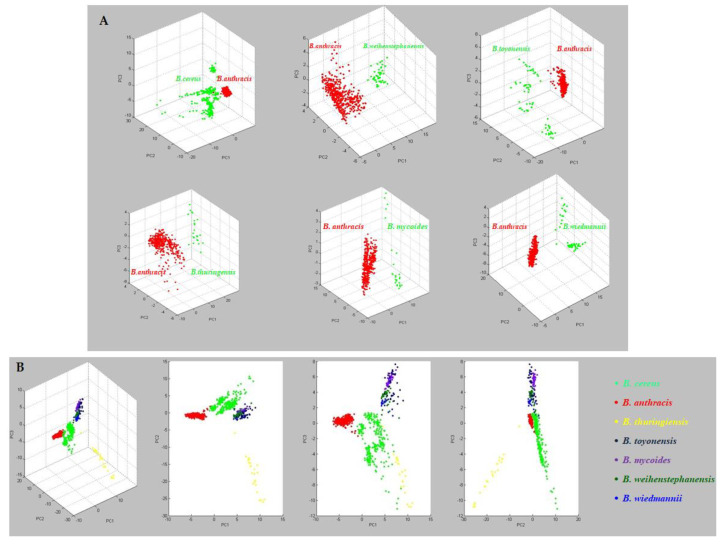
(**A**) Statistical comparison by PCA between mass spectra of *B. anthracis* and of other species belonging to the *B. cereus* group. (**B**) PCA grouped the analysed isolates into 7 different clusters.

**Figure 4 microorganisms-09-01202-f004:**
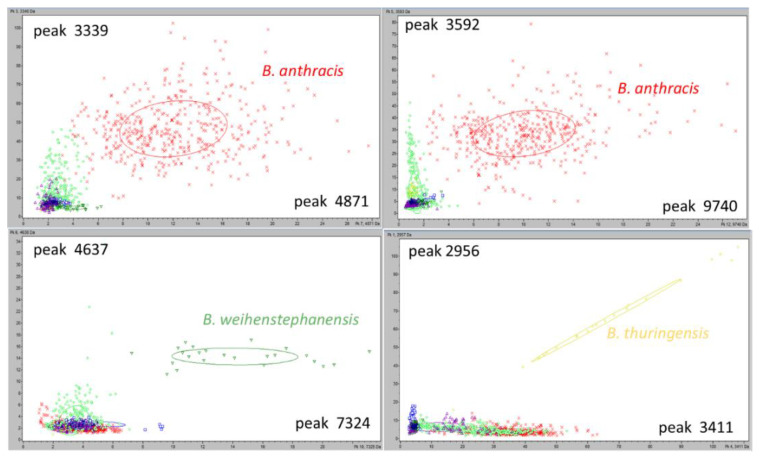
Two-dimensional scatter plot of characteristic ion peaks for *B. anthracis*, B. *weihentephanensis* and B. *thuringensis*. The characteristic peaks served as the x- and y-axes, respectively. The intensities of the characteristic ion peaks were expressed in arbitrary intensity units. The ellipses represent the 95% confidence intervals of peak intensities for each species.

**Figure 5 microorganisms-09-01202-f005:**
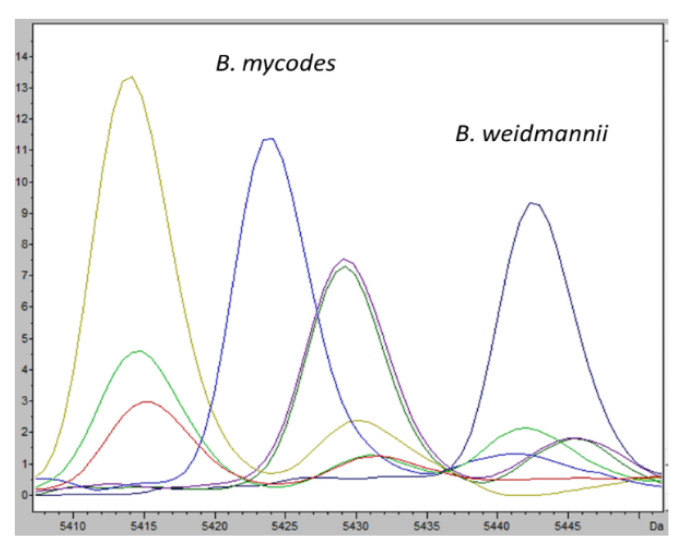
Average mass spectra of characteristic ion peaks among various *Bacillus* species understudy. Intensities of characteristic ion peaks (*m*/*z* 5424 for *B. mycoides* in light blue, *m*/*z* 5443 for *B. weidmannii* in dark blue) are expressed in arbitrary intensity units.

**Figure 6 microorganisms-09-01202-f006:**
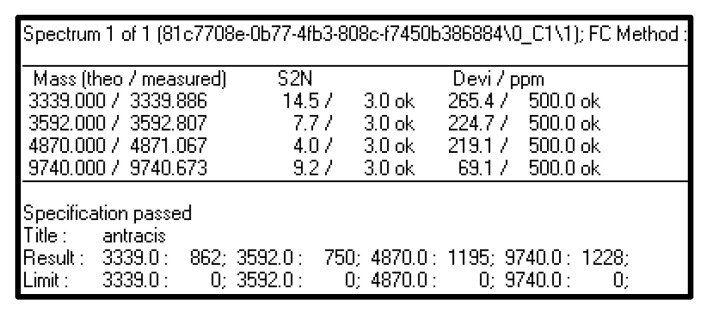
Result of FlexAnalysis algorithm for *B. anthracis* species. If all the characteristic ion peaks are found in the experimental spectrum considered, the specification is passed and the correct and unambiguous identification at species level is possible.

**Figure 7 microorganisms-09-01202-f007:**
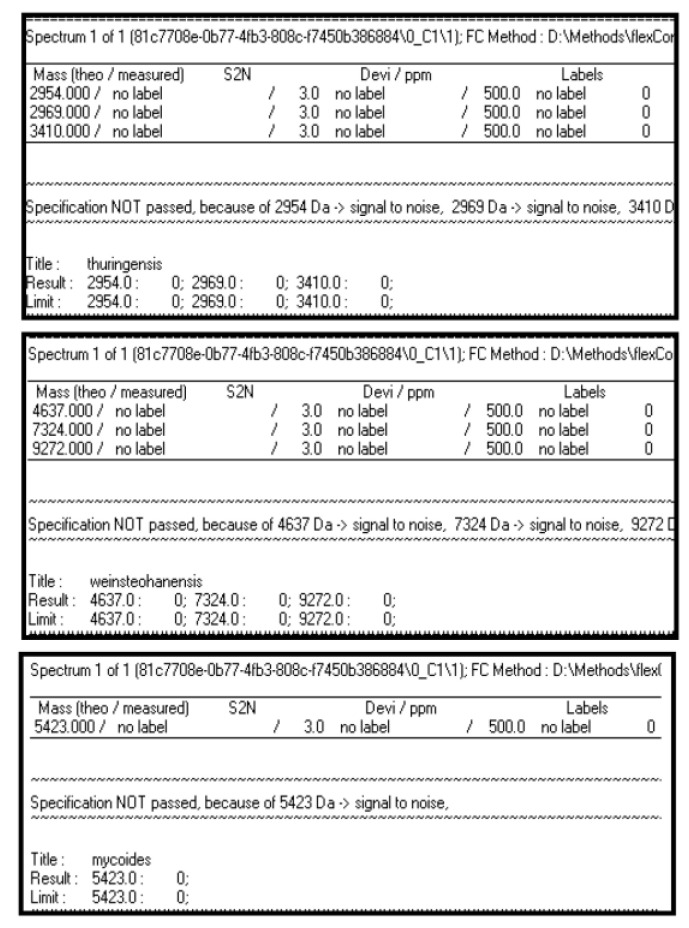
Results of FlexAnalysis algorithms for all the considered species, applied to an experimental spectrum of *B. anthracis* strain.

**Table 1 microorganisms-09-01202-t001:** List of the 12 ion peaks considered with their mass to charge ratio and the peak intensity average for each class and relative standard deviation. In bold are shown the higher intensities values for each mass peak, to be considered as characteristic ion peak for the class/species.

Intensity ± Stdev							
Mass/Charge	*B. anthracis*	*B. cereus*	*B. mycoides*	*B. thuringensis*	*B. toyonensis*	*B. weihenstephanensis*	*B. weidmannii*
2956	2.32 ± 0.72	3.23 ± 0.98	8.79 ± 2.84	**46.91 ± 16.43**	3.78 ± 1.13	3.97 ± 0.69	4.83 ± 0.59
2968	1.8 ± 0.48	2 ± 0.6	3.35 ± 0.87	**46.04 ± 15.53**	2.59 ± 0.47	2.29 ± 0.41	3 ± 0.36
3339	**33.75 ± 10.78**	8.52 ± 6.22	5.43 ± 0.91	3.78 ± 0.65	4.98 ± 1.32	3.63 ± 0.68	5.62 ± 0.81
3411	6.72 ± 5.97	5.48 ± 7.98	2.82 ± 0.80	**47.62 ± 17.3**	8.65 ± 6.48	3.36 ± 1.2	3.3 ± 0.66
3592	**24.26 ± 7.01**	6.07 ± 6.07	3.94 ± o.82	8 ± 1.56	3.67 ± 0.75	3.75 ± 0.81	3.63 ± 0.5
4637	1.47 ± 0.45	2.25 ± 1.77	1.94 ± 0.43	1.27 ± 0.29	1.56 ± 0.32	**10.35 ± 1.04**	1.85 ± 0.35
4871	**8.71 ± 3.21**	2.01 ± 0.53	2.03 ± 0.44	1.44 ± 0.33	1.58 ± 0.31	2.87 ± 0.74	1.64 ± 0.27
5424	1.56 ± 0.87	1.94 ± 1.52	**12.71 ± 1.63**	2.57 ± 0.78	6.52 ± 3.8	6.18 ± 1.54	1.33 ± 0.29
5443	1.53 ± 0.43	3.87 ± 4.16	2.35 ± 0.44	1.58 ± 0.37	2.91 ± 0.96	2.76 ± 0.68	**10.33 ± 1.73**
7324	2.99 ± 0.84	2.6 ± 0.79	3.42 ± 1.63	2.07 ± 0.27	2.64 ± 0.64	**10.23 ± 3.05**	2.77 ± 0.45
9272	0.62 ± 0.24	0.96 ± 0.23	0.88 ± 0.15	0.59 ± 0.14	0.79 ± 0.19	**4.26 ± 0.62**	0.89 ± 0.20
9740	**7.29 ± 2.97**	0.98 ± 0.41	1.26 ± 0.39	0.8 ± 0.16	0.74 ± 0.24	1.4 ± 0.4	0.78 ± 0.15

**Table 2 microorganisms-09-01202-t002:** Results of ClinProtools statistical model quick classifier (QC) and Support Vector Machine (SVM) in terms of Recognition Capability and Cross Validation for overall and single *Bacillus* species considered.

		overall	*B. anthracis*	*B. cereus*	*B. mycoides*	*B. thuringensis*	*B. toyonensis*	*B. weihentephanensis*	*B. wiedmannii*
QC	Recognition Capability	68.3%	91.95%	25.97%	81.48%	95.45%	35.71%	75%	72.5%
	Cross Validation	66.59%	92.35%	24.85%	78.72%	91.89%	29.91%	72.73%	75.68%
SVM	Recognition Capability	97.14%	100%	98.79%	96.3%	95.45%	91.96%	100%	97.5%
	Cross Validation	96.02%	100%	97.54%	97.92%	89.19%	92.86%	100%	94.67%

## Data Availability

The data presented in this study are available on request from the corresponding author.

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
