# Peer review of "Discrimination of *Bacillus cereus* Group Members by MALDI-TOF Mass Spectrometry"

_microorganisms, 2021, doi:10.3390/microorganisms9061202_

Round 1
Reviewer 1 Report
The aim of this study was to identify B. cereus group strains to identify specific biomarkers in the spectra of each species which would allow them to be unequivocally identified. First paragraph of the abstract is very general and it is already known information. It is better described in abstract more specifically information about study.
Line 91 Listeria must be with italic. Line 99-101 it is better to write the number of species without n. or you can add n=. In part, material and method is better described from which food were strains isolated. Horizontal method after isolates is not necessary.
Line 125 and is without italics. Line 166-172 check which is with italic and which words not. Line 209-211 name of Bacillus must be with italics.
Figure 2 is not very good quality. Why didn't the authors upload a dendrogram from MALDI -TOF?
In manuscript there are very many formal mistakes. Discussion needs description in this in this state. It is insufficient.
Conclusion is a separate chapter. Supplementary file is in format which can not be opened.
Author Response
Reviewer 1
Dear Reviewer,
Thank you for your valuable comments and for helping us to improve our manuscript.
The aim of this study was to identify B. cereus group strains to identify specific biomarkers in the spectra of each species which would allow them to be unequivocally identified. First paragraph of the abstract is very general and it is already known information. It is better described in abstract more specifically information about study.
The abstract should be a total of about 200 words maximum for this reason the information that we can provide are limited. In any case, I added some information as requested.
Line 91. Listeria must be with italic.
I wrote Listeria in italics.
Line 99-101. It is better to write the number of species without n. or you can add n=.
I replaced n. with n = at lines 99-101.
In part, material and method is better described from which food were strains isolated. Horizontal method after isolates is not necessary.
I removed the sentence “Microbiology of food and animal feeding stuffs – Horizontal method for the determination of low numbers of presumptive Bacillus cereus – Most probable number technique and detection method”.
Line 125 and is without italics.
I removed the italics from the word “and” at line 125.
Line 166-172 check which is with italic and which words not.
I removed the italics from some words at lines 166-172.
Line 209-211 name of Bacillus must be with italics.
I added italics to the name of Bacillus at lines 209-211.
Figure 2 is not very good quality.
For Figure 2, it is not possible to increase the resolution. We will ask the journal to help us with this problem.
Why didn't the authors upload a dendrogram from MALDI -TOF?
The dendrogram was loaded as a supplementary file.
In manuscript there are very many formal mistakes. Discussion needs description in this in this state. It is insufficient. Conclusion is a separate chapter. Supplementary file is in format which cannot be opened.
I tried to improve the manuscript and correct some errors. I combined the results with discussion in a section called "Results and Discussion", as asked by the other reviewer, and created a new section called "Conclusions". I uploaded the supplementary files again.
Reviewer 2 Report
Topic fits well under Aims and scope of Microorganisms journal – Special Issue "Matrix-Assisted Laser Desorption/Ionization Time of Flight Mass Spectrometry (MALDI-TOF MS) for the Identification of Pathogenic Microorganisms". Topic of this study is interesting and results/conclusions may potentially be interesting to broad audience of journal readers, including from food industry, clinical microbiology, instrument/software developers field and researcher.
In general, the authors should improve the presentation of their results, the results must be clearly presented and discussed and methods adequately described. English must be improved. I suggest corrections by a native speaker – especially results and discussion sections! In addition, I kindly ask you to consider my comments/suggestions, improve your manuscript and increase your chances for being accepted for publication:
- Lines 95-96: “In addition, will be extended the identification also to other species of Bacillus cereus group as B. toyonensis and B. wiedmannii, not included now in the BDAL library.”
My comment: Please rewrite the sentence. (correct English)
- Introduction is good, however, extremely important information and literature references are missing, for example on the MALDI TOF MS application in B.cereus identification. Please search the literature and REWRITE the introduction: give an overview of what was done and published in the “specific” field (“MALDI TOF MS application in B.cereus identification”) prior your experiments, and emphasize what makes your research and results different/new, what is a crucial progress that you present here for the field. In a fast search I already found several relevant and important publications that were not mentioned in your manuscript:
https://www.frontiersin.org/articles/10.3389/fmicb.2020.511674/full
https://pubmed.ncbi.nlm.nih.gov/31216842/
https://www.nature.com/articles/srep16989
https://aem.asm.org/content/75/22/7229
https://journals.plos.org/plosone/article?id=10.1371/journal.pone.0217078
- Lines 72-85- My comment: These sentences can be deleted from the Introduction or may be moved into discussion.
- My comment: When abbreviations are used for the first time a complete word/name must be provided as well. Please correct these, for example in line 17 (MALDI- TOF MS), line 90 (MBT), line 96 (BDAL)…
- Line 96: BDAL library
My comment: This is a commercial library, right!? Please provide vendor's (producer's) name.
- Line 89-91: “This method is already used by the additional software module MBT Compass Subtyping for the correct species identification for Listeria.
My comment: A literature reference is missing here. Please provide it.
- Lines 91-94. “The Listeria species have very similar spectra profile, and the classical identification approach allows the genus identification but a new algorithm developed by Bruker Daltonik are able to identify correctly the species using few characteristic and known peaks.”
My comment: A literature reference is missing here. Please provide it.
- In line 86 you mention “In this study, a collection of 160 strains belonging to the Bacillus cereus group….) and in line 99 “Bacterial strains used in this study included n. 103 strains isolated from food samples….”
My comment: Where from did the other strains come from? Please enlarge the sentence in lines 103-104 “The isolates used in this study are listed in Table S1.” Into “The complete list of isolates used in this study and their origin are listed in Table S1. Also in the main text and abstract please put the number of samples per each strain.
- Lines 119-120: “Bacterial DNA was extracted using the DNAeasy Blood and Tissue kit (Qiagen, Hilden, Germany), according to the protocol for Gram-positive bacteria.”
My comment: Please provide the protocol or relevant literature reference.
- Lines 125-126: “For the identification of B. cereus s.s., B. mycoides, B. thuringiensis and B. weihenstephanensis, were used two essays combined…”
My comment: Please correct English.
- Lines 135-136: “….MALDI-TOF analysis.”
My comment: Correct into “…MALDI-TOF MS analysis.”
- Section 2.2.
My comment producers of solvents and regents are missing, for example for TFA (line 137), water (line 142), HCCA (line 145), etc. Please provide their names.
- Section 2.2.
My comment: From presented it seems that samples were prepared only once and multiple mass spectra (18 in total) were obtained from one spot. Did you evaluate technical reproducibility – sample preparation is extremely important for MALDI TOF MS fingerprinting and the quality of MS data? Please provide data which proves the high quality of your generated MALDI TOF MS data?
- Line 152: “…spectra…”
My comment: Please correct into “…mass spectra…” Please correct the latter in whole manuscript, e.g. also in lines 180, 229, etc….
- Lines 164 -176 – My comment: These sentences fit better into Results/Discussion part. Please move them.
- Line 178: “…by the full extraction protocol procedures.”
My comment: Please provide literature reference or full description of the latter protocol.
- Line 178: “Each extract was spotted on eight spots…”
My comment: Please provide additional explanation how the latter was performed. Information is missing.
- Line 179. “…three different mass spectra were acquired for each of them.”
My comment: Please provide additional explanation how the latter was performed. Information is missing.
- Lines 180 - 181: “…for species were analyzed by Flex Analysis software to verify the accuracy in terms of mass to charge ratio and the reproducibility.”
My comment: Please provide additional explanation how the latter was performed. How did you evaluate this? Information is missing.
- Lines 181- 182: “After the spectra quality check the new MSP’s (main spectra profile) for B. toyonensis and B. weidmannii…”
My comment: Please provide additional explanation, how did you perform the quality check? Information is missing.
- Lines 183-184: “The new MSP’s were tested preparing in the following days several spot starting from sample subculture and performing the identification.”
My comment: Please correct – it is unclear what you try to say. If additional samples were prepared and analysed this should be explained in more detail.
- Lines 186 – 187: “All the experimental spectra for all the species will be processed together in ClinProTools software (v. 3.0, Bruker Daltonik GmbH, Bremen, Germany) for statistical analysis.
My comment: Please correct English. (will!?)
- Lines 196-198:” In this study, as expected, the identification with commercial databases proved to be inconclusive for the B. cereus group species because almost all samples mass spectra matched with B. anthracis (in the SR), with a log(score) >2.0 in most cases.”
My comment: where is this visible? Cross-referencing in the text is needed. Please include.
- Lines 209-210: MY comment: Please use italic.
- Section 3. Results
My comment: I strongly suggest merging of Results and Discussion sections into one section (results and discussion). The way the results and discussion sections are written at the moment, it is difficult to read and understand what you mean to say (please correct English, rewrite sentences! You have for every sentence a new paragraph!), you discuss results in the Results only section. Please rewrite the two sections. Also, do not repeat and explain gain the execution of experiments in the Results and Discussion section, instead use cross-referencing if needed. In addition, where did you present the results for the experiments explained in section 2.1. ? This is missing. Cross-referencing in the text is needed. Please include.
- Lines 218-219: “…each spectrum has been subjected to spectral preprocessing procedures, such as smoothing, baseline subtraction, and intensity normalization.”
My comment: This should be mentioned in Materials and methods section, and please provide more details on how the latter was performed (processing parameters are missing).
- Line 230: “…for each species a main spectrum (MSP) has been created…”
My comment: How was MSP created? The answer should be provided in the Materials and methods section.
- Line 232: “In the dendrogram generated by MSPs….”
My comment: How was a dendrogram created? What data was used and what software?
I strongly suggest you must INSERT a new subtitle 2.2.1 Data processing and statistical interpretation – where you must give ALL details on how the MS data was processed. MATERIALS AND METHODS SECTION must be written with ALL details and information (MS data processing and statistical part is missing both) – When all is provided and when your manuscript is published then readers can repeat your experiments with the same outcome.
- Lines 227-228: “The spectra demonstrate a relatively high signal-to-noise ratio, which typically permits the detection of 50 to 100 mass peaks per spectrum.”
My comment: How high was signal-to-noise ratio this must be provided.
- Line 241: “…generation of simulated gel views”
My comment: This should be mentioned in Materials and methods section, and please provide more details on how the latter was generated.
- My comment: Figures 1 and 2 are of low resolution, please provide figures of better quality. Figure 3. – instead of using light green use another color.
- Line 250: “…using Principal Component Analysis (PCA).”
My comment: This should be mentioned in Materials and methods section, and please provide more details on how PCA was done, what parameters were used, what kind of data (intensity, m/z values, or both, in what mass range) was analysed, etc.
- Lines 259 – 276 – My comment: These sentences belong in the Materials and methods section.
- My comment: It is unclear how table 1 was created. You do mention parameters that were taken into account (in lines 271- 275), but not the boundaries for each of enumerated parameters. Please provide details how did you eliminate many of 100 peaks and end up with only 12 ion peaks.
- Lines 276, 280, etc. and table 1. My comment: Please use “ion peaks” in stead “peaks”.
- Lines 283- 284: “For B. anthracis, B. thuringensis and B. weinsthephanensis species, more than one characteristic peak was defined.”
My comment: Please be precise, and provide answers in the manuscript: how many of ion peaks and which precisely (add additional table) were defined as unique to each strain.
- In figure 4. My comment: It is clear that for every distinction two ion peaks are responsible, right? This figure should be discussed properly.
- Lines 306- 308: “Classification models were generated using the algorithms QC (Quick Classifiers) and SVM (Support Vector Machine) available in the software ClinPro-tools software (v. 3.0, Bruker Daltonik GmbH, Bremen, Germany
My comment: This sentence belong in the Materials and methods section.
- Table 2. My comment: It is unclear how table 2 was created – please provide explanation in Materials and methods section.
- Lines 346-247: “The algorithms in FlexAnalysis software (v. 3.4., Bruker Daltonik GmbH, Bremen, Germany) were created…”
My comment: How did you create these algorithms, please provide detailed explanation in the manuscript in the separate Materials and methods sub-section.
- Line 362. – My comment: Please create a separate section called “Conclusions”
- My comment: In discussion, please provide a detailed comparison between already published methods and yours experimental approach. Please provide an explanation: what makes your approach more advanced compared to what was done and published already, and why researches should use your methods instead of the other methods in the literature. In general, discussion part, as presented now, is POOR and it should be enlarged, by incorporating detailed discussion of presented results (figures 1-5, tables 1 and 2 and supplementary) and by comparison of your results with previously published data.
- Graphical abstract is missing. Please provide.
My final decision is RECONSIDERATION after MAJOR revision.
Author Response
Reviewer 2
Dear Reviewer,
Thank you for your valuable comments and for helping us to improve our manuscript.
Topic fits well under Aims and scope of Microorganisms journal – Special Issue "Matrix-Assisted Laser Desorption/Ionization Time of Flight Mass Spectrometry (MALDI-TOF MS) for the Identification of Pathogenic Microorganisms". Topic of this study is interesting and results/conclusions may potentially be interesting to broad audience of journal readers, including from food industry, clinical microbiology, instrument/software developers’ field and researcher.
In general, the authors should improve the presentation of their results, the results must be clearly presented and discussed and methods adequately described. English must be improved. I suggest corrections by a native speaker – especially results and discussion sections! In addition, I kindly ask you to consider my comments/suggestions, improve your manuscript and increase your chances for being accepted for publication:
- Lines 95-96: “In addition, will be extended the identification also to other species of Bacillus cereus group as B. toyonensis and B. wiedmannii, not included now in the BDAL library.”
My comment: Please rewrite the sentence. (correct English)
The sentence was reworded as requested.
- Introduction is good, however, extremely important information and literature references are missing, for example on the MALDI TOF MS application in B.cereus identification. Please search the literature and REWRITE the introduction: give an overview of what was done and published in the “specific” field (“MALDI TOF MS application in B.cereus identification”) prior your experiments, and emphasize what makes your research and results different/new, what is a crucial progress that you present here for the field. In a fast search I already found several relevant and important publications that were not mentioned in your manuscript:
https://www.frontiersin.org/articles/10.3389/fmicb.2020.511674/full
https://pubmed.ncbi.nlm.nih.gov/31216842/
https://www.nature.com/articles/srep16989
https://aem.asm.org/content/75/22/7229
https://journals.plos.org/plosone/article?id=10.1371/journal.pone.0217078
Thanks for the comment.
As suggested, I added in the introduction some literature data related to similar studies.
In fact, our study adds to those already available in the literature. Similarly to Lash et al., 2015 and Pauker at al., 2018, we have created new libraries with the numerous experimental spectra obtained from the analysis of 160 strains belonging to the B.cereus group, which are added to the spectra of commercial libraries, in order to increase the probability of a secure identification.
The novelty of this study, compared to the studies already conducted, is extension our study to other species of the Bacillus cereus group such as B. toyonensis and B. wiedmannii, as they are currently not included in the BDAL library (Bruker Daltonik GmbH, Bremen, Germany). Furthermore, we created algorithms to unequivocally identify species after the classic library matching approach. These innovations are now explained in the introduction.
- Lines 72-85- My comment: These sentences can be deleted from the Introduction or may be moved into discussion.
The sentence at the lines 72-85 has moved to the “Conclusion”.
- My comment: When abbreviations are used for the first time a complete word/name must be provided as well. Please correct these, for example in line 17 (MALDI- TOF MS), line 90 (MBT), line 96 (BDAL)…
I added the full name to the first citation of all abbreviations.
- Line 96: BDAL library
My comment: This is a commercial library, right!? Please provide vendor's (producer's) name.
I added the producer at the line 96.
- Line 89-91: “This method is already used by the additional software module MBT Compass Subtyping for the correct species identification for Listeria.
My comment: A literature reference is missing here. Please provide it.
I added the literature reference.
- Lines 91-94. “The Listeria species have very similar spectra profile, and the classical identification approach allows the genus identification but a new algorithm developed by Bruker Daltonik are able to identify correctly the species using few characteristic and known peaks.”
My comment: A literature reference is missing here. Please provide it.
I added the literature reference.
- In line 86 you mention “In this study, a collection of 160 strains belonging to the Bacillus cereus group….) and in line 99 “Bacterial strains used in this study included n. 103 strains isolated from food samples….”
My comment: Where from did the other strains come from? Please enlarge the sentence in lines 103-104 “The isolates used in this study are listed in Table S1.” Into “The complete list of isolates used in this study and their origin are listed in Table S1. Also, in the main text and abstract please put the number of samples per each strain.
I changed the sentence "The isolates used in this study are listed in Table S1" into "The complete list of isolates used in this study and their origin are listed in Table S1" at the lines 103-104. We added the number of analyzed strains in the abstract.
- Lines 119-120: “Bacterial DNA was extracted using the DNAeasy Blood and Tissue kit (Qiagen, Hilden, Germany), according to the protocol for Gram-positive bacteria.”
This is a classic protocol that can be found on the specific data sheet.
- Lines 125-126: “For the identification of B. cereus s.s., B. mycoides, B. thuringiensis and B. weihenstephanensis, were used two essays combined…”
My comment: Please correct English.
I corrected English.
- Lines 135-136: “….MALDI-TOF analysis.”
My comment: Correct into “…MALDI-TOF MS analysis.”
I corrected the sentence at the line 135-136.
- Section 2.2.
My comment producers of solvents and regents are missing, for example for TFA (line 137), water (line 142), HCCA (line 145), etc. Please provide their names.
I added the producers of solvents and reagents used at the lines 137, 142, 145.
- Section 2.2.
My comment: From presented it seems that samples were prepared only once and multiple mass spectra (18 in total) were obtained from one spot. Did you evaluate technical reproducibility – sample preparation is extremely important for MALDI TOF MS fingerprinting and the quality of MS data? Please provide data which proves the high quality of your generated MALDI TOF MS data?
Thanks for the comment. Sample preparation was performed three times, in three different moments. After each preparation, the sample was applied to 18 different wells of the 96-well steel target plate. The 18 mass spectra acquired for each strain were compared to check for the presence of flatline spectra or spectra with anomalous peaks. Furthermore, the mass accuracy interspect was evaluated by verifying the mass displacement of the base peak in all the acquired mass spectra, considering the maximum tolerance 500 ppm.
- Line 152: “…spectra…”
My comment: Please correct into “…mass spectra…” Please correct the latter in whole manuscript, e.g. also in lines 180, 229, etc….
I added “mass” spectra throughout the manuscript.
- Lines 164 -176 – My comment: These sentences fit better into Results/Discussion part. Please move them.
I moved these sentences to the “Result and Discussion” section.
- Line 178: “…by the full extraction protocol procedures.”
My comment: Please provide literature reference or full description of the latter protocol.
I added the full description of the protocol.
- Line 178: “Each extract was spotted on eight spots…”
My comment: Please provide additional explanation how the latter was performed. Information is missing. ?????
I added the missing information.
- Line 179. “…three different mass spectra were acquired for each of them.”
My comment: Please provide additional explanation how the latter was performed. Information is missing. ?????
I added the missing information.
- Lines 180 - 181: “…for species were analyzed by Flex Analysis software to verify the accuracy in terms of mass to charge ratio and the reproducibility.”
My comment: Please provide additional explanation how the latter was performed. How did you evaluate this? Information is missing.
I added the missing information.
- Lines 181- 182: “After the spectra quality check the new MSP’s (main spectra profile) for B. toyonensis and B. weidmannii…”
My comment: Please provide additional explanation, how did you perform the quality check? Information is missing.
I added the missing information.
- Lines 183-184: “The new MSP’s were tested preparing in the following days several spot starting from sample subculture and performing the identification.”
My comment: Please correct – it is unclear what you try to say. If additional samples were prepared and analysed this should be explained in more detail.
I added the missing information and reworded the sentence.
- Lines 186 – 187: “All the experimental spectra for all the species will be processed together in ClinProTools software (v. 3.0, Bruker Daltonik GmbH, Bremen, Germany) for statistical analysis.
My comment: Please correct English. (will!?)
It was a typo. I replaced with “were”.
- Lines 196-198:” In this study, as expected, the identification with commercial databases proved to be inconclusive for the B. cereus group species because almost all samples mass spectra matched with B. anthracis (in the SR), with a log(score) >2.0 in most cases.”
My comment: where is this visible? Cross-referencing in the text is needed. Please include. ????
Unfortunately, is difficult to describe what is stated in lines 196-198. The only evidence that shows this point is the report generated by the MBT Compass 4.1. software. Hereafter are some visual examples.
Please see attachment for the strains.
- Lines 209-210: MY comment: Please use italic.
I added italics at the lines 209-210.
- Section 3. Results
My comment: I strongly suggest merging of Results and Discussion sections into one section (results and discussion). The way the results and discussion sections are written at the moment, it is difficult to read and understand what you mean to say (please correct English, rewrite sentences! You have for every sentence a new paragraph!), you discuss results in the Results only section. Please rewrite the two sections. Also, do not repeat and explain gain the execution of experiments in the Results and Discussion section, instead use cross-referencing if needed. In addition, where did you present the results for the experiments explained in section 2.1.? This is missing. Cross-referencing in the text is needed. Please include.
As recommended, I merged the results with discussion in the new section called "Results and Discussion". The strains used in our study, whose identification is the result of the biomolecular analyses of the section 2.1, are indicated in table S1. If considered necessary, I could include the results of individual PCR.
- Lines 218-219: “…each spectrum has been subjected to spectral preprocessing procedures, such as smoothing, baseline subtraction, and intensity normalization.”
My comment: This should be mentioned in Materials and methods section, and please provide more details on how the latter was performed (processing parameters are missing).
I moved this sentence in Materials and methods section and added the references of the software.
- Line 230: “…for each species a main spectrum (MSP) has been created…”
My comment: How was MSP created? The answer should be provided in the Materials and methods section.
I described as MSP were created in the Materials and methods section.
- Line 232: “In the dendrogram generated by MSPs….”
My comment: How was a dendrogram created? What data was used and what software?
I added the software by which the dendrogram was created.
I strongly suggest you must INSERT a new subtitle 2.2.1 Data processing and statistical interpretation – where you must give ALL details on how the MS data was processed. MATERIALS AND METHODS SECTION must be written with ALL details and information (MS data processing and statistical part is missing both) – When all is provided and when your manuscript is published then readers can repeat your experiments with the same outcome.
In the "Materials and Methods" I created a new section called "Data processing".
- Lines 227-228: “The spectra demonstrate a relatively high signal-to-noise ratio, which typically permits the detection of 50 to 100 mass peaks per spectrum.”
My comment: How high was signal-to-noise ratio this must be provided.
I changed the sentence into "The mass spectra demonstrate a relatively high signal-to-noise ratio, which typically permits the detection of 50 to 100 mass peaks per spectrum with the signal to noise ratio higher than 3" at the lines 227-228.
- Line 241: “…generation of simulated gel views”
My comment: This should be mentioned in Materials and methods section, and please provide more details on how the latter was generated.
I moved this sentence in Materials and methods section.
- My comment: Figures 1 and 2 are of low resolution, please provide figures of better quality. Figure 3. – instead of using light green use another color.
Thanks for the comment. We would have liked to change the color but unfortunately the software does not allow the change. Light green is the default, we cannot change it. I made the inscription “B. cereus” more visible in Fig. 3B.
For Figures 1 and 2, is not possible to increase the resolution. We will ask the journal to help us with this.
- Line 250: “…using Principal Component Analysis (PCA).”
My comment: This should be mentioned in Materials and methods section, and please provide more details on how PCA was done, what parameters were used, what kind of data (intensity, m/z values, or both, in what mass range) was analysed, etc.
I added the missing information in Materials and methods section.
- Lines 259 – 276 – My comment: These sentences belong in the Materials and methods section.
The sentence at the lines 259 – 276 has moved to Materials and methods section.
- My comment: It is unclear how table 1 was created. You do mention parameters that were considered (in lines 271- 275), but not the boundaries for each of enumerated parameters. Please provide details how did you eliminate many of 100 peaks and end up with only 12 ion peaks.
“To reduce the data to a shorter list, only the mass peaks with higher Dave, and the lower standard deviation for the peak intensity average of the main class were considered”. This sentence was inserted in the manuscript.
- Lines 276, 280, etc. and table 1. My comment: Please use “ion peaks” instead “peaks”.
I added “ion” peaks in all the manuscript.
- Lines 283- 284: “For B. anthracis, B. thuringensis and B. weinsthephanensis species, more than one characteristic peak was defined.”
My comment: Please be precise, and provide answers in the manuscript: how many of ion peaks and which precisely (add additional table) were defined as unique to each strain.
I added the characteristic peaks for each strain in the manuscript as required.
- In figure 4. My comment: It is clear that for every distinction two ion peaks are responsible, right? This figure should be discussed properly.
I added a more detailed caption in relation to Figure 4.
- Lines 306- 308: “Classification models were generated using the algorithms QC (Quick Classifiers) and SVM (Support Vector Machine) available in the software ClinPro-tools software (v. 3.0, Bruker Daltonik GmbH, Bremen, Germany
My comment: This sentence belongs in the Materials and methods section.
I moved this sentence to Materials and methods section.
- Table 2. My comment: It is unclear how table 2 was created – please provide explanation in Materials and methods section.
I added the missing information in Materials and methods section.
- Lines 346-247: “The algorithms in FlexAnalysis software (v. 3.4., Bruker Daltonik GmbH, Bremen, Germany) were created…”
My comment: How did you create these algorithms, please provide detailed explanation in the manuscript in the separate Materials and methods sub-section.
I added more details about the algorithms. I created a new section called "Data processing".
- Line 362. – My comment: Please create a separate section called “Conclusions”
As recommended, I combined the results with discussion in a section called "Results and Discussion" and created a new section called "Conclusions".
- My comment: In discussion, please provide a detailed comparison between already published methods and your experimental approach. Please provide an explanation: what makes your approach more advanced compared to what was done and published already, and why researches should use your methods instead of the other methods in the literature. In general, discussion part, as presented now, is POOR and it should be enlarged, by incorporating detailed discussion of presented results (figures 1-5, tables 1 and 2 and supplementary) and by comparison of your results with previously published data.
Thanks for your comment. I have created, as required, a section called "Results and Discussion" in which the results of this study and the figures included in the manuscript are well explained. Our study adds scientific evidence to previous studies already conducted in this field with the aim of unequivocally discriminating the members of the B.cereus group under study, with particular attention to B.anthracis. Following your suggestions, I tried to improve the manuscript.
- Graphical abstract is missing. Please provide ??????
I didn’t provide it because is not obligatorily required by the journal.
Valeria Rondinone

Round 2
Reviewer 1 Report
Author made regular changes, but the conclusion is not with references. Conclusion summarized the best results of study.
Author Response
Dear Reviewer,
Thanks for your suggestions.
I added others references in the conclusions, as requested.
Best regards,
Valeria Rondinone.
Reviewer 2 Report
Kindly thank for improvements.
Author Response
Dear Reviewer,
Thank you again for your valuable comments and for helping us to improve our manuscript.
Best regards,
Valeria Rondinone.